# Predictive Factors Associated with Future Decline in Swallowing Function among Japanese Older People Aged ≥ 75 Years

**DOI:** 10.3390/ijerph21060674

**Published:** 2024-05-24

**Authors:** Komei Iwai, Tetsuji Azuma, Takatoshi Yonenaga, Yasuyuki Sasai, Yoshinari Komatsu, Koichiro Tabata, Taketsugu Nomura, Iwane Sugiura, Yujo Inagawa, Yusuke Matsumoto, Seiji Nakashima, Yoshikazu Abe, Takaaki Tomofuji

**Affiliations:** 1Department of Community Oral Health, School of Dentistry, Asahi University, 1-1851 Hozumi, Mizuho 501-0296, Gifu, Japan; ko-mei@dent.asahi-u.ac.jp (K.I.); tetsuji@dent.asahi-u.ac.jp (T.A.); yone0730@dent.asahi-u.ac.jp (T.Y.); ysssi0718@dent.asahi-u.ac.jp (Y.S.); km2-km2@dent.asahi-u.ac.jp (Y.K.); tabata-k@dent.asahi-u.ac.jp (K.T.); 2Gifu Dental Association, 1-18 Minamidori, Kano-cho, Gifu 500-8486, Gifu, Japan; takezou@ma.ctk.ne.jp (T.N.); white@rhythm.ocn.ne.jp (I.S.); yujo@alato.ne.jp (Y.I.); whitebear@crest.ocn.ne.jp (Y.M.); semsynakashi@shopping3.gmobb.jp (S.N.); kjkcg552@ybb.ne.jp (Y.A.)

**Keywords:** biting, choking, dry mouth, periodontal pocket depth, swallowing function, longitudinal study

## Abstract

Predictive factors associated with a decline in swallowing function after 2 years were examined in 3409 Japanese older people aged ≥ 75 years who had undergone a dental checkup in Gifu Prefecture, Japan. Participants with normal swallowing function in a baseline survey in April 2018 were followed for 2 years. Swallowing function was assessed using a repetitive saliva swallowing test. In our study, 429 participants (13%) who were swallowing less than three times in 30 s based on a repetitive saliva swallowing test after 2 years were diagnosed as those with decline in swallowing function. Multivariate logistic regression analyses showed the decline in swallowing function after 2 years was associated with the male gender (odds ratio [ORs]: 0.772; 95% confidence interval [CIs]: 0.615–0.969), age ≥ 81 years (presence; ORs: 1.523; 95% CIs: 1.224–1.895), support/care-need certification (presence; ORs: 1.815; 95% CIs: 1.361–2.394), periodontal pocket depth (PPD) ≥ 4 mm (presence; ORs: 1.469; 95% CIs: 1.163–1.856), difficulty in biting hard food (yes; ORs: 1.439; 95% CIs: 1.145–1.808), choking on tea and water (yes; ORs: 2.543; 95% CIs: 2.025–3.193), and dry mouth (yes; ORs: 1.316; 95% CIs: 1.052–1.646) at baseline. Therefore, the dental checkup items associated with a decline in swallowing function after 2 years were a PPD ≥ 4 mm, difficulty in biting hard food, choking on tea and water, and dry mouth. PPD status and confirming to the self-administered questionnaire about biting, choking, and dry mouth may be useful in predicting future decline in swallowing function.

## 1. Introduction

A decline in swallowing function is a condition in which a disorder occurs somewhere in the series of swallowing processes that causes aspiration, choking, and dehydration [1]. In older people, a long-term decline in swallowing function can easily lead to malnutrition, and consequently, a reduced quality of life (QOL) [2]. In Japan, the proportion of older people with a decline in swallowing function has been increasing in recent years, being found in 27.2% of older people in 2020 [3]. In addition, a decline in swallowing function is often irreversible, making it difficult to return to a normal state, even with treatment [4]. Therefore, the early detection and control of factors associated with a decline in swallowing function are important for maintaining QOL and are of public health significance.

Previous studies have reported a relationship between oral status and swallowing function. For example, decreased salivation could contribute to impaired swallowing function [5,6]. It is also reported that a negative correlation existed between the number of present teeth and swallowing function [7]. Furthermore, it is known that impaired swallowing function is associated with a number of *Fusobacterium* spp., one of the bacteria associated with periodontal disease [8]. Based on these studies, it is suggested that oral health is associated with swallowing function.

The determinant of decline in swallowing function was reported to be due to the loss of muscle mass with aging [9,10]. In addition, stress, depression, and other psychiatric disorders were also reported as the risk of decline in swallowing function [11,12]. On the other hand, it is still unclear which oral factors contribute to a future decline in swallowing function in older people.

Dental checkups in Gifu, Japan (*Gifu sawayaka koku kenshin*) are conducted once a year for Japanese older people aged ≥ 75 years [13,14]. This checkup also includes content related to swallowing function. Therefore, a longitudinal study of these data may make it possible to search for factors that predict a future decline in swallowing function. In the present study, we hypothesized that some dental checkup items might be associated with a future decline in swallowing function. Therefore, we conducted a longitudinal study over a 2-year period with the aim of clarifying the longitudinal associations between dental checkup items and a decline in swallowing function in Japanese older people aged ≥ 75 years.

## 2. Materials and Methods

### 2.1. Participants

Data were analyzed from local residents who had undergone a dental checkup in Gifu city, Kagamihara city, Kani city, and Ogaki city in Gifu Prefecture, Japan. Between April 2018 and March 2019, a total of 8584 Japanese older people aged ≥ 75 years participated in the baseline survey. Participants with dementia (n = 450) and those showing a decline in swallowing function at baseline (those who could not swallow at least three times in 30 s in a repetitive saliva swallowing test [12]; n = 974) were excluded. Participants with missing data on swallowing function (n = 47) and body mass index (BMI) (n = 2975) were also excluded from the analysis. Of the remaining 4138 participants, 3409 were followed from April 2020 to March 2021 (follow-up rate: 82%). Therefore, data from 3409 community-dwelling residents (1428 males and 1981 females; mean age at baseline: 81 years) were analyzed in the present study (Figure 1).

### 2.2. Checkup Items by Dentists

Data on the following dental checkup items by dentists were collected: swallowing function, presence or absence of ≥20 present teeth, presence or absence of decayed teeth, and presence or absence of a periodontal pocket depth (PPD) ≥ 4 mm. In this checkup, we did not use other additional methods such as X-rays. The data for the dental checkup items were provided by the Gifu Dental Association, Japan. Swallowing function was assessed using a repetitive saliva swallowing test [15]. Participants who were swallowing less than three times in 30 s based on the repetitive saliva swallowing test after 2 years were diagnosed as those with decline in swallowing function [15]. The coded values of the Community Periodontal Index were used to evaluate PPD ≥ 4 mm, with codes 1 and 2 being evaluated as PPD ≥ 4 mm [16].

### 2.3. Self-Administered Questionnaire Items

Data on difficulty in biting hard food, choking on tea and water, dry mouth, and smoking habits were provided by the Gifu Dental Association, Japan. On the self-administered questionnaire, the participants were asked to choose “yes” or “no” for the following items: “I have difficulty biting hard food”, “I choke on tea and water”, and “I have dry mouth” [17]. Regarding smoking habits, participants who smoked at least one cigarette per day were included (presence or absence) [18]. In our study, we used the original questionnaire commonly used in dental checkups in Gifu, Japan. Furthermore, the validity and reproducibility of these questionnaire items (chewing, choking, dry mouth, and smoking habits) used in our study was reported in previous studies [19,20,21].

### 2.4. Survey Items in the National Health Insurance Database System

Data on gender, age, BMI, and presence or absence of hypertension, diabetes, dyslipidemia, musculoskeletal disorders, and support/care-need certification were obtained from the National Health Insurance Database of Japan (NDB) [22,23].

### 2.5. Statistical Analysis

Differences in the participants’ characteristics at baseline and after 2 years were assessed using Fisher’s exact test. Univariate and multivariate logistic regression analyses were performed using the presence of a decline in swallowing function as the dependent variable. In the multivariate stepwise logistic regression analysis, in addition to gender and age, the variables that were significantly different in the univariate logistic regression analysis were selected as confounders and set as adjustment variables. We confirmed the suitability of this model using the Hosmer–Lemeshow fit test. These data were then analyzed using SPSS statistical analysis software (version 27; IBM Japan, Tokyo, Japan). All *p*-values < 0.05 were considered significant.

### 2.6. Ethics

This study was approved by the Ethics Review Committee of Asahi University (No. 33006) and performed in accordance with the Declaration of Helsinki (as revised in Brazil 2013).

## 3. Results

Table 1 shows the characteristics of the participants at baseline and after 2 years. The proportions of participants with hypertension (*p* = 0.009), diabetes (*p* = 0.008), musculoskeletal disorders (*p* = 0.005), and support/care-need certification (*p* < 0.001) were significantly higher after 2 years than at baseline. In addition, the proportion of participants with ≥20 present teeth were significantly lower after 2 years than at baseline (*p* < 0.001). However, no significant differences in decayed teeth, PPD ≥ 4 mm, difficulty in biting hard food, choking on tea and water, and dry mouth were found between at baseline and after 2 years.

Table 2 shows the crude odds ratios [ORs] and 95% confidence intervals [CIs] for a decline in swallowing function after 2 years. In the present study, 429 participants (13%) were newly diagnosed with a decline in swallowing function after 2 years. The results indicated that the risk of a decline in swallowing function after 2 years was significantly correlated with the male gender (ORs: 0.711; 95% CIs: 0.550–0.918), age ≥ 81 years (presence; ORs: 1.834; 95% CIs: 1.493–2.251), musculoskeletal disorders (presence; ORs: 1.419; 95% CIs: 1.102–1.828), support/care-need certification (presence; ORs: 2.743; 95% CIs: 2.121–3.546), decayed teeth (presence; ORs: 1.319; 95% CIs: 1.058–1.645), PPD ≥ 4 mm (presence; ORs: 1.424; 95% CIs: 1.137–1.784), difficulty in biting hard food (yes; ORs: 1.907; 95% CIs: 1.538–2.365), choking on tea and water (yes; ORs: 3.020; 95% CIs: 2.436–3.744), and dry mouth (yes; ORs: 1.844; 95% CIs: 1.497–2.272) at baseline.

Table 3 shows the adjusted ORs and 95% CIs for a decline in swallowing function after 2 years. After adjusting for gender, age, musculoskeletal disorders, support/care-need certification, decayed teeth, PPD, difficulty in biting hard food, choking on tea and water, and dry mouth, the risk of decline in swallowing function after 2 years was significantly correlated with the male gender (ORs: 0.772; 95% CIs: 0.615–0.969), age ≥ 81 years (presence; ORs: 1.523; 95% CIs: 1.224–1.895), support/care-need certification (presence; ORs: 1.815; 95% CIs: 1.361–2.394), PPD ≥ 4 mm (presence; ORs: 1.469; 95% CIs: 1.163–1.856), difficulty in biting hard food (yes; ORs: 1.439; 95% CIs: 1.145–1.808), choking on tea and water (yes; ORs: 2.543; 95% CIs: 2.025–3.193), and dry mouth (yes; ORs: 1.316; 95% CIs: 1.052–1.646) at baseline.

## 4. Discussion

To the best of our knowledge, this is the first longitudinal study to examine the associations between dental checkup items and a decline in swallowing function in Japanese older people. The results of a logistic regression analysis showed that, after adjusting for gender, age, musculoskeletal disorders, support/care-need certification, decayed teeth, PPD, difficulty in biting hard food, choking on tea and water, and dry mouth, the presence or absence of a decline in swallowing function after 2 years was associated with PPD ≥ 4 mm, difficulty in biting hard food, choking on tea and water, and dry mouth at baseline. From these results, having PPD ≥ 4 mm, difficulty in biting hard food, choking on tea and water, and dry mouth were predicted to be associated with a higher risk of a decline in swallowing function in the future.

Previous studies have reported a relationship between a decline in swallowing function and overall health, showing that dysphagia was associated with diabetes and the development of dementia [24,25]. A decline in swallowing function has also been reported to be associated with malnutrition and future mortality [26]. These reports indicate that a decline in swallowing function could be detrimental to overall health. In addition, once swallowing function declines, it is difficult to return to the original state [4]. Therefore, early screening to prevent a decline in swallowing function through dental checkups is important for the maintenance and promotion of overall health. However, early dental interventions may be necessary, especially for those with factors found to be associated with a decline in swallowing function during dental checkups.

In our study, there was an association between the selection of difficulty in biting hard food, choking on tea and water, or dry mouth in a self-administered questionnaire and a decline in swallowing function. People with difficulty biting hard food seemed to have a reduce chewing ability and cerebral blood flow [27,28]. Decreased cerebral blood flow interferes with proper swallowing motor regulation [29]. Therefore, the participants with difficulty in biting hard food may have experienced a decrease in swallowing function in our study. In addition, it is reported that people who choke easily could have decreased function of the infrahyoid muscles [30], making it difficult to elevate the hyoid bone and larynx, resulting in a decline in swallowing function [31]. Furthermore, since decreased salivary secretion and dry mouth is caused by stress and age-related atrophy of salivary gland acinar cells [32,33], people with dry mouth would experience difficulty in forming and transporting food boluses at mealtime, resulting in a decline in swallowing function [6,34]. These previous observations are consistent with the present results, which indicated that difficulty in biting hard food, choking on tea and water, and dry mouth were associated with a decline in future swallowing function according to a self-administered questionnaire using data from dental checkups. On the other hand, the present diagnoses were self-reported and not based on specialized knowledge or instruments. In other words, these factors can be noticed by one’s own senses, regardless of whether one has undergone a dental checkup. The results of the present study suggest that if a person is aware of their difficulty with chewing hard foods, choking on tea or water, or dry mouth, they should undergo a dental checkup or visit a dental clinic to help prevent a decline in swallowing function.

The results of the present study indicated that a PPD ≥ 4 mm is a risk factor for a decline in swallowing function. A previous epidemiological study reported that approximately 56% of older patients with a decline in swallowing function had periodontal disease [35]. Another study reported that people with a decline in swallowing function possess greater amounts of periodontal disease-associated bacteria (such as *Fusobacterium* spp.) than do those with normal swallowing function [8]. The present study and these previous studies are consistent, in that they all found that periodontal disease is associated with a decline in swallowing function.

In Japan, based on the Long-Term Care Insurance Law, people who require daily and continuous support and care because of various disabilities can receive support/care-need certification [36]. This certification can then be classified into seven types, starting with those with minor illnesses, namely, support-need certifications 1–2 and care-need certifications 1–5 [36]. The results of the present study indicated that support/care-need certification is a risk factor for a decline in swallowing function. This observation is in agreement with previous studies reporting that a significantly higher proportion of people with support/care-need certification had a decline in swallowing function compared with those without support/care-need certification [37,38]. Therefore, the present and previous studies support the need for measures to prevent a decline in swallowing function precisely for those with support/care-need certification.

In our study, there was no association between decline in swallowing function and <20 present teeth or decayed teeth. Previous studies have reported that dental caries and a low number of present teeth decrease all oral functions [39,40], including swallowing function, which differs from the results of our study. This may be related to the proportion of the target population. According to Survey of Dental Diseases, in 2022, the proportion of persons with ≥20 present teeth at age 80 years was 51% (67% in our study), and the proportion of persons with decayed teeth was 35% (26% in our study) [41]. It is possible that the results of our study differed from the results of previous studies because the oral health of the population in our study was better than that of the average population in Japan. Therefore, the results of our study should be considered for external validity.

In the present study, we evaluated multivariate logistic regression analysis models using the Hosmer–Lemeshow fit test. The Hosmer–Lemeshow fit test is used to examine the fit of a multivariate logistic regression analysis model and tests whether the observed event rate in a subgroup model fits the expected event rate. The Hosmer–Lemeshow test is considered to fit well with *p*-values > 0.05 [42]. In the present study, the *p*-value was 0.432, suggesting that it fits well.

However, this study has several limitations. First, because the participants underwent regular dental checkups (at least once in both 2018 and 2020), they may have been a more health-conscious sample than the general population. Therefore, if different health populations are targeted, the results may differ. Second, because the NDB was used, the presence or absence of diseases and disorders not in the database were unknown. Thirdly, we used a self-administered questionnaire in our study. In the future, we will consider adding objective evaluations in addition to the self-administered questionnaire items to increase the reliability of our observations. Fourth, we could not confirm detailed information on the names of the medications that the participants were taking. We could not determine the presence or absence of medications affecting salivary secretion and therefore did not include medications in our analysis. Finally, although it is reported that ENT (ear, nose, and throat) and depressive states reflect the risk of decline in swallowing function in older people [12,43], we could not collect these data based on dental checkups. On the other hand, a major strength of this study is that it was a longitudinal study of more than 3400 Japanese older people aged ≥ 75 years, which is useful for establishing causal associations with a decline in swallowing function. Furthermore, it was possible to gather study population data from multiple areas in Japan (Gifu city, Kagamihara city, Kani city, and Ogaki city).

## 5. Conclusions

The results of the present study indicate that among the dental checkup items, PPD ≥ 4 mm, difficulty in biting hard food, choking on tea and water, and dry mouth are associated with a higher risk for a future decline in swallowing function among Japanese older people aged ≥ 75 years. PPD status and confirming to the self-administered questionnaire about biting, choking, and dry mouth may be useful in predicting future decline in swallowing function.

## Figures and Tables

**Figure 1 ijerph-21-00674-f001:**
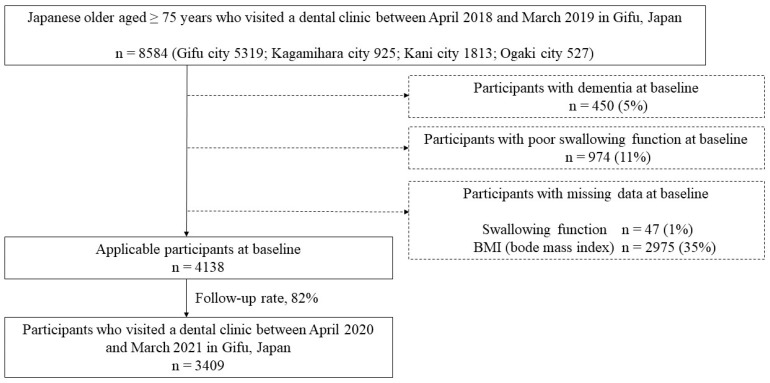
Flowchart of the data selection criteria.

**Table 1 ijerph-21-00674-t001:** Participants’ characteristics (n = 3409).

Factor	Baseline (% **)	After 2 Years (% **)	*p*-Value *
Gender ^†^	1428 (42%)	1428 (42%)	-
Age (years)			
75–80	1851 (54%)	761 (22%)	-
81-	1558 (46%)	2648 (78%)	
BMI (kg/m^2^)			
<18.5	265 (8%)	227 (7%)	0.095
≥18.5	3144 (92%)	3182 (93%)	
Smoking habits ^‡^	34 (1%)	35 (1%)	0.635
Hypertension ^‡^	2077 (61%)	2182 (64%)	0.009
Diabetes ^‡^	1193 (35%)	1299 (38%)	0.008
Dyslipidemia ^‡^	1803 (53%)	1858 (55%)	0.182
Musculoskeletal disorders ^‡^	2571 (75%)	2668 (78%)	0.005
Support/care-need certification ^‡^	383 (11%)	763 (22%)	<0.001
Number of present teeth (tooth)			
−19	1142 (33%)	1303 (38%)	<0.001
20-	2267 (67%)	2106 (62%)	
Decayed teeth ^‡^	888 (26%)	836 (25%)	0.147
Periodontal pockets (mm)			
<4	1152 (34%)	1088 (32%)	0.099
≥4	2257 (66%)	2321 (68%)	
Difficulty in biting hard food ^§^	830 (24%)	851 (25%)	0.555
Choking on tea and water ^§^	700 (21%)	743 (22%)	0.202
Dry mouth ^§^	1019 (30%)	1094 (32%)	0.050

Abbreviations: BMI, body mass index. * *p* < 0.05, using Fishers exact test. ** Percentage is rounded to one decimal place. ^†^: Male (proportion of male); ^‡^: presence (proportion of presence); ^§^: yes (proportion of yes).

**Table 2 ijerph-21-00674-t002:** Crude ORs and 95% CIs for a decline in swallowing function at follow-up.

Factor		ORs	95% CIs	*p*-Value
Gender	Female	1	(reference)	0.009
	Male	0.711	0.550–0.918	
Age (years)	75–80	1	(reference)	<0.001
	81-	1.834	1.493–2.251	
BMI (kg/m^2^)	<18.5	1	(reference)	0.146
	≥18.5	0.771	0.544–1.094	
Smoking habits	Absence	1	(reference)	0.506
	Presence	0.668	0.203–2.195	
Hypertension	Absence	1	(reference)	0.244
	Presence	1.133	0.918–1.397	
Diabetes	Absence	1	(reference)	0.704
	Presence	1.042	0.844–1.287	
Dyslipidemia	Absence	1	(reference)	0.790
	Presence	1.028	0.839–1.259	
Musculoskeletal disorders	Absence	1	(reference)	0.007
	Presence	1.419	1.102–1.828	
Support/care-need certification	Absence	1	(reference)	<0.001
	Presence	2.743	2.121–3.546	
Number of present teeth (tooth)	−19	1	(reference)	0.050
	20-	0.811	0.658–1.000	
Decayed teeth	Absence	1	(reference)	0.014
	Presence	1.319	1.058–1.645	
Periodontal pockets (mm)	<4	1	(reference)	0.002
	≥4	1.424	1.137–1.784	
Difficulty in biting hard food	Yes	1	(reference)	<0.001
	No	1.907	1.538–2.365	
Choking on tea and water	Yes	1	(reference)	<0.001
	No	3.020	2.436–3.744	
Dry mouth	Yes	1	(reference)	<0.001
	No	1.844	1.497–2.272	

Abbreviations: ORs, odds ratios; CIs, confidence interval; BMI, body mass index.

**Table 3 ijerph-21-00674-t003:** Adjusted ORs and 95% CIs for a decline in swallowing function at follow-up.

Factor		ORs	95% CIs	*p*-Value
Gender	Female	1	(reference)	0.025
	Male	0.772	0.615–0.969	
Age	75–80	1	(reference)	<0.001
	81-	1.523	1.224–1.895	
Musculoskeletal disorders	Absence	1	(reference)	0.420
	Presence	1.117	0.853–1.463	
Support/care-need certification	Absence	1	(reference)	<0.001
	Presence	1.815	1.361–2.394	
Decayed teeth	Absence	1	(reference)	0.154
	Presence	1.184	0.938–1.494	
Periodontal pockets (mm)	<4	1	(reference)	0.001
	≥4	1.469	1.163–1.856	
Difficulty in biting hard food	No	1	(reference)	0.002
	Yes	1.439	1.145–1.808	
Choking on tea and water	No	1	(reference)	<0.001
	Yes	2.543	2.025–3.193	
Dry mouth	No	1	(reference)	0.016
	Yes	1.316	1.052–1.646	

Abbreviations: ORs, odds ratios; CIs, confidence interval. Adjustment for gender, age, musculoskeletal disorders, support/care-need certification, decayed teeth, periodontal pockets, difficulty in biting hard food, choking on tea and water, dry mouth. Hosmer–Lemeshow fit test: *p* = 0.432.

## Data Availability

The data that support the findings of this study are available from the Gifu National Health Insurance Federation and Wide-Area Federation of Medical Care for Late-Stage Older People, but restrictions apply to the availability of these data, which were used under license for the present study and are not publicly available. However, the data are available from the authors upon reasonable request and with permission from the Gifu National Health Insurance Federation and Wide-Area Federation of Medical Care for Late-Stage Older People.

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
