# Peer review of "Predictive Factors Associated with Future Decline in Swallowing Function among Japanese Older People Aged ≥ 75 Years"

_ijerph, 2024, doi:10.3390/ijerph21060674_

Round 1

Reviewer 1 Report

Comments and Suggestions for Authors

Respected Authors,

I find your paper entitled " Predictive Factors Associated with Future Decline in Swallowing Function among Japanese Older People Aged ≥ 75 Years " interesting and I consider that it addresses a very important issue for the quality of life of the elderly.

However, I consider that it needs some improvements. Please find below my observations and recommendations.

In the Introduction section:

-lines 46-47: "impaired swallowing function has been indicated as a potential contributing factor to decreased salivation " - isn't it vice versa?

-please add more details concerning the relationship between oral health status and swallowing function, as well as details concerning the possible determinants of swallowing function impairment.

In the Materials and Methods section:

-more details concerning oral health status assessment are necessary: what instruments were used? Was it based on clinical exam only, or were any other additional methods used (such as X-rays, for instance)?

-more details concerning the questionnaire used are necessary: was it an original questionnaire, or it was translated from another language? Either way, it requires validation for the target population, and details of the validation process should be provided

-was salivary secretion evaluated only on the basis of the subjective sensation of dry mouth? An objective evaluation method would have been more useful, because the decrease in salivary secretion is not always accompanied by the sensation of dry mouth, as there is also the possibility that the sensation of dry mouth is accompanied by a normal functioning of the salivary glands (pseudoxerostomia or spurious xerostomia) 

-an assessment of the medication taken by the subjects would have been useful, not only at baseline, but throughout the study process, as medication can be changed during the study, some medication may be added that reduces salivary secretion.

In the Results section:

-Table 1 - please add "N (%)" for number and percentage in the first row, under "Baseline" and "After 2 years".

In the Discussion section:

-more comments should be added upon the obtained results, not only PPD, certification, and dry mouth sensation. It is a known fact that reduced salivary flow rate is followed by the impairment of swallowing (this is one of the functions of saliva), but the cause of the decrease in saliva secretion and how to intervene to remedy it are important

-more studies should be included in this section. It is mentioned "previous studies", but only one (19) is cited, or "decreased salivation causes a decline in swallowing function" and only reference [20] is cited. There are many more studies in the specialized literature on this topic

-please add comments upon the other factors considered in this study, which did not prove to be predictors, but are important (number of present teeth, dental caries).

The study is important because it underlines the importance of maintaining good oral health, but comments should be added concerning other factors that could determine swallowing impairment in the elderly (including ENT factors, or depressive states etc.)

In the References section:

-more references should be included, the selected theme is very vast.

Author Response

-lines 46-47: "impaired swallowing function has been indicated as a potential contributing factor to decreased salivation " - isn't it vice versa?

Response: We thank the reviewer for your valuable advice. As you point out, decreased salivation contribute to impaired swallowing function. We have revised the text. (lines 47-48)

-please add more details concerning the relationship between oral health status and swallowing function, as well as details concerning the possible determinants of swallowing function impairment.

Response: We thank the reviewer for your valuable advice. We have added the sentences about the relationship between oral health status and swallowing function, and the determinants of swallowing function impairment. (lines 46-57)

-more details concerning oral health status assessment are necessary: what instruments were used? Was it based on clinical exam only, or were any other additional methods used (such as X-rays, for instance)?

Response: We thank the reviewer for your valuable advice. In our study, the assessment of oral health status was based on clinical exam only. Therefore, we did not use other additional methods such as X-rays. We have emphasized this point to avoid confusing to the reader. (lines 87-88)

-more details concerning the questionnaire used are necessary: was it an original questionnaire, or it was translated from another language? Either way, it requires validation for the target population, and details of the validation process should be provided.

Response: We thank the reviewer for your valuable advice. We used the original questionnaire commonly used in dental checkup in Gifu, Japan. Furthermore, the validity and reproducibility of regarding this questionnaire items (chewing, choking, and dry mouth, and smoking habits) used in our study was reported in previous studies. We have added the sentences and “references” according to your suggestion. (lines 101-104, References 19-21)

-was salivary secretion evaluated only on the basis of the subjective sensation of dry mouth? An objective evaluation method would have been more useful, because the decrease in salivary secretion is not always accompanied by the sensation of dry mouth, as there is also the possibility that the sensation of dry mouth is accompanied by a normal functioning of the salivary glands (pseudoxerostomia or spurious xerostomia).

Response: We thank the reviewer for your valuable advice. In our study, evaluation of dry mouth is based on subjective sensations only from a self-administered questionnaire. The validity and reproducibility of this questionnaire dry mouth item was reported in past studies. However, as you pointed out, adding objective evaluations in addition to subjective items would increase the reliability of our observations. This point will be an issue for future consideration. We have added the limitation in the text. (lines 242-245)

-an assessment of the medication taken by the subjects would have been useful, not only at baseline, but throughout the study process, as medication can be changed during the study, some medication may be added that reduces salivary secretion.

Response: We thank the reviewer for your valuable advice. We could confirm whether and how many medications participants were taking, but do not possess detailed information on the names of the medications. We could not determine the presence or absence of medications affecting salivary secretion and therefore did not include medications in our analysis. We have emphasized this limitation in the text. (lines 245-248)

-Table 1 - please add "N (%)" for number and percentage in the first row, under "Baseline" and "After 2 years".

Response: We thank the reviewer for your valuable advice. We added % to the first row in Table 1 according to your suggestions. (Table 1)

-more comments should be added upon the obtained results, not only PPD, certification, and dry mouth sensation. It is a known fact that reduced salivary flow rate is followed by the impairment of swallowing (this is one of the functions of saliva), but the cause of the decrease in saliva secretion and how to intervene to remedy it are important.

Response: We thank the reviewer for your valuable advice. We have added the discussion of the results other than PPD, certification, and dry mouth according to your suggestions. (lines 182-193)

We have also added the sentence of the cause and intervene of decrease in saliva secretion. (lines 190-191)

-more studies should be included in this section. It is mentioned "previous studies", but only one (19) is cited, or "decreased salivation causes a decline in swallowing function" and only reference [20] is cited. There are many more studies in the specialized literature on this topic.

Response: We thank the reviewer for your valuable advice. We revised the text in discussion and added the references. (lines 182-202, References 28-34)

The references number changed from 19 to 27. We also changed the development of the argument of discussion and removed previous references 20.

-please add comments upon the other factors considered in this study, which did not prove to be predictors, but are important (number of present teeth, dental caries). The study is important because it underlines the importance of maintaining good oral health, but comments should be added concerning other factors that could determine swallowing impairment in the elderly (including ENT factors, or depressive states etc.).

Response: We thank the reviewer for your valuable advice. In our study, there was no association between decline in swallowing function and ≥ 20 present teeth or decayed teeth, which differs from the results of past study. This may be related to the proportion of the target population. According to Survey of Dental Diseases in 2022, the proportion of persons with ≥ 20 present teeth at age 80 years was 51% (67% in our study), and the proportion of persons with decayed teeth was 35% (26% in our study). It is possible that the results of our study differed from the results of previous studies because the oral health of the population in our study was better than that of the average population in Japan. Therefore, the results of our study should be considered for external validity. We have added the sentences. (lines 222-231)

Furthermore, as you point out, ENT (Ear, Nose, and Throat) and depressive states are the cause of poor swallowing function in the elderly. However, we do not possess this information. Therefore, we have added the sentence as limitations. (lines 248-250, References 43, 12)

-more references should be included, the selected theme is very vast.

Response: We thank the reviewer for your valuable advice. We revised “references” appropriately.

Reviewer 2 Report

Comments and Suggestions for Authors

 Summary: Manuscript ijerph-3001646 is a longitudinal study (2 years follow-up) on risk factors associated with dysphagia in older adults. The study was carried out in Japan and surveyed more than 3,400 individuals (out of an initial 8,500 individuals), and found an association between a decline in swallowing function and several parameters of general health (e.g., musculoskeletal disorders) and dental health (e.g., number of present teeth) status. This is an important finding as dysphagia is a cause of other health problems and even deaths in the elderly. Some of the health parameters surveyed can be improved or prevented by proper health care, so the reported results are of significance to the field of Public Health. There are however a few points that I could not understand from the current manuscript, and therefore need some clarification.

General (major) comment: How did the authors define “decline in swallowing function”? Three different questions were asked in relation to dysphagia. Was a yes response to any of the three questions on follow-up considered a decline, or did they use some kind of score? The only assessment of swallowing function was these three yes/no questions, so the authors should provide a more detailed description of how the answers differed between initial and follow-up assessments. This is also important in order to understand the odds ratio statistics.

Minor comments:

Introduction, line 46: “...impaired swallowing function has been indicated as a potential contributing factor to decreased salivation” -- Is it not the other way around? Decreased salivation contributes to impaired swallowing.

Table 1 - add in the legend that the values in % have been rounded off.

Discussion, line 209: “suggesting that it fit well.” Change “fit” to “fits”

Author Response

-How did the authors define “decline in swallowing function”? Three different questions were asked in relation to dysphagia. Was a yes response to any of the three questions on follow-up considered a decline, or did they use some kind of score? The only assessment of swallowing function was these three yes/no questions, so the authors should provide a more detailed description of how the answers differed between initial and follow-up assessments. This is also important in order to understand the odds ratio statistics.

Response: We thank the reviewer for your valuable advice. We do not use a self-administered questionnaires to assess swallowing function. Swallowing function was assessed by a repetitive saliva swallowing test. Participants with normal swallowing function in a baseline were followed, and participants who were swallowing less than three times in 30 seconds by a repetitive saliva swallowing test after 2 years were diagnosed as those with decline in swallowing function in our study. We have revised the text. (lines 18-21, 90-92)

-line 46: “...impaired swallowing function has been indicated as a potential contributing factor to decreased salivation” -- Is it not the other way around? Decreased salivation contributes to impaired swallowing.

Response: We thank the reviewer for your valuable advice. As you pointed out, decreased salivation can contribute to impaired swallowing function. We have revised the text. (lines 47-48)

-Table 1 - add in the legend that the values in % have been rounded off.

Response: We thank the reviewer for your valuable advice. We have added the text regarding percentages are rounded to one decimal place to “Table 1, Legend”. (Table 1)

-line 209: “suggesting that it fit well.” Change “fit” to “fits”

Response: We thank the reviewer for your valuable advice. We have revised the sentence from “fit” to “fits” according to your suggestion. (line 237)

Round 2

Reviewer 1 Report

Comments and Suggestions for Authors

Respected Authors,

Thank you for considering my recommendations in the improved version of your paper.

Please have the paper checked for English correction.

Comments on the Quality of English Language

The paper needs moderate English correction.